# Fish Oil Replacement by Camelina (*Camelina sativa* L.) Oil in Diets for Juvenile Tench (*Tinca tinca* L.): Effects on Survival, Growth, and Whole-Body Fatty Acid Profile

**DOI:** 10.3390/ani12233362

**Published:** 2022-11-30

**Authors:** Teresa García, José M. Carral, María Sáez-Royuela, Jesús D. Celada

**Affiliations:** Departamento de Producción Animal, Facultad de Veterinaria, Universidad de León, Campus de Vegazana s/n, 24071 León, Spain

**Keywords:** juvenile tench, camelina oil, growth, fatty acid profile, nutritional indices

## Abstract

**Simple Summary:**

The total replacement of fish oil (FO) with camelina oil (CO) in the diets of juvenile tench (*Tinca tinca* L.) did not have negative effects on survival rates or growth performance. When juveniles were fed with diets containing levels of CO of 40% or higher, lipid content in whole-body tench was significantly lower than those fed the control diet whereas linolenic acid content was significantly higher. No differences in total saturated, monounsaturated, or polyunsaturated fatty acids were found in whole-body juveniles. The nutritional indices were within optimal values for healthy foods for human consumption.

**Abstract:**

Tench (*Tinca tinca* L.) plays a key role in the diversification of inland aquaculture, but its culture is mainly based on extensive culture systems with usually low and unpredictable yields. Rearing procedures under controlled conditions are essential to promote and consolidate tench production, and it is necessary to set up adequate feeding in early growth phases. Fish oil (FO) is currently the main source of lipids in aquafeeds, but considering the stagnation of smaller pelagic fisheries, alternative oils should be addressed. In a ninety-day experiment, the effects of partial and total replacement of FO with camelina oil (CO) on juvenile tench growth performance and whole-body composition were evaluated. Six isonitrogenous and isolipidic diets with different levels of CO were tested: 0% (control), 20%, 40%, 60%, 80%, and 100%. The survival rate was 100%, and no significant differences in growth performance (total length, weight, specific growth rate, feed conversion ratio, and biomass gain) were found. The lipid content in whole-body juveniles was significantly lower when juveniles were fed diets containing 40% and higher levels of CO than those fed the control diet whereas linolenic acid content was significantly higher. No differences in ΣSFA, ΣMUFA, ΣPUFA, Σn − 3, or Σn − 6 whole-body content were found. The nutritional indices ΣPUFA/ΣSFA and Σn − 6/Σn − 3 showed a linear increase trend with dietary CO inclusion whereas the EPA + DHA showed an opposite tendency. Compared to the control diet, EPA + DHA content (g kg^−1^) was significantly lower in juvenile tench fed a 100% CO diet, and Σn − 6/Σn − 3 was significantly higher in juvenile tench fed 80% and 100% CO diets. Overall, the results indicate that the total replacement of FO with CO in diets is feasible without negative effects on growth performance whereas the nutritional quality of juvenile tench was unaffected with a maximum replacement of 80%.

## 1. Introduction

According to OECD/FAO [1], the increase of aquatic animals for human consumption is driven primarily by the continued growth of aquaculture production, which is expected to reach 108 Mt by 2031. This predicted growth must comply with one of the new priority areas of the Strategic Framework of FAO for 2022–2031, the Blue Transformation, which focuses on the development of more efficient, inclusive, resilient, and sustainable aquaculture systems [2]. At present, this sector is the biggest consumer of fishmeal (FM) and fish oil (FO) with a global use in 2019 of 78% and 68%, respectively [3], and thus the environmental footprint derived from their use in feedstuffs is the main concern to reach a sustainable growth [4]. The estimation of FO production by 2031 is 1.3 Mt, of which 53% will come from foraged species [1]. Considering the low efficiency of FO production, one ton of pelagic fish is needed to obtain an average of 50 kg of oil [5], and high and undesirable fishing pressure is expected. For this reason, research to find alternative lipid sources is driven, given that oils from terrestrial plants are the largest and most widely used since, in general terms, vegetable oil (VO) production is more stable [6] and prices are lower than those for FO [7]. In fact, animal feed formulations include increased amounts of VO, especially for aquaculture [1]. A large variety of plant oils, alone or in blends, have been considered FO substitutes for many aquaculture species, highlighting soybean, palm, linseed, sunflower, rapeseed, and olive oils [6,8]. Lipids from vegetable sources are usually rich in C18 fatty acids, mainly linoleic (LA, 18:2n − 6), α-linolenic (ALA, 18:3n − 3), and oleic (OA, 18:1n − 9), but they lack or have a very limited content of long-chain polyunsaturated n − 3 fatty acids (n − 3 LC-PUFA), such as eicosapentaenoic (EPA, 20:5n − 3) and docosahexaenoic (DHA, 22:6n − 3) acids.

A potentially new source of n − 3 PUFA is the Brassicaceae species *Camelina sativa* (commonly known as ‘gold-of-pleasure’ or ‘false flax’). Interest in its production has increased in recent years due not only to its agrotechnical benefits, which make it more environmentally friendly than other conventional oilseed crops [9,10], but also due to the fatty acid profile of its oil. Zanetti et al. [11] summarized a considerable number of studies on camelina’s adaptability to broad environmental conditions, low requirements for water and nutrients, resistance to multiple insect pests and diseases as well as its multiple uses in food, feed, and biobased applications. In relation to feed uses, camelina oil (CO) has a high content of PUFA’s (>55%); the n − 3 LC-PUFA precursor ALA and values of n − 3/n − 6 ratio close to two stand out because they are uncommon in VO [12,13,14]. According to Tocher [15], the partial or full substitution of FO with VO is more feasible for freshwater fish than marine species, which apparently lack the ability to desaturate and elongate C-18 PUFAs and, therefore, are more likely to have n − 3 LC-PUFA deficiency. Negative effects on growth performance when FO was partially or totally replaced by CO have been reported in gilthead sea bream (*Sparus aurata*) by Huyben et al [16] and Ofori-Mensah et al. [17], Atlantic cod (*Gadus morhua*) by Hixon and Parrish [18], and totoaba (*Totoaba macdonaldi*) by Maldonado-Othon et al. [19]. However, total FO replacement by CO did not affect the growth performance of freshwater species, such as rainbow trout (*Oncorhynchus mykiss*) [20], Nile tilapia (*Oreochromis niloticus*) [21], and the diadromous Atlantic salmon (*Salmo salar*) [22,23,24,25]. In search of alternative sources of EPA and DHA, oil from genetically modified *C. sativa* expressing algal genes was tested in European sea bass (Dicentrarchus labrax) [26], gilthead sea bream [16,27], and Atlantic salmon [23,24]. The inclusion of oil from transgenic camelina did not result in negative effects on fish performance or nutritional attributes of farmed fish.

Final fish composition as whole-body or fish fillet usually reflects feed composition. When CO was included in the diets, a reduction in EPA and DHA content was reported [16,17,19,21,22,23,28,29]. For this reason, it is necessary to evaluate the changes in the fatty acid profile in fish and thus if diets including CO compromise the healthy benefits associated with human consumption of fish.

According to Sicuro [30], around 73% of aquatic-farmed species are so-called ‘minor species’ because they are responsible for 1% of total aquaculture production but play a relevant role as a diversity reservoir, as is the case of tench (*Tinca tinca* L.). This cyprinid is highly appreciated in many European countries mainly for its gastronomic quality but also as an attractive sport-fishing species [31,32]. Tench production is based on extensive culture systems where growth is often limited, leading to usually low and unpredictable yields [32,33]. The development of intensification techniques is necessary to promote and consolidate tench production, and establishing rearing procedures under controlled conditions is essential to set up adequate feeding in early growth phases. As a specific diet is essential to face nutritional studies, González-Rodríguez et al. [34] and García et al. [31] proposed a practical diet for juvenile tench which allowed for good juvenile survival and growth performance. Later, Sáez-Royuela et al. [35] determined that an 8.5% dietary lipid content (2% of cod liver oil) should be present in the formulation. Recently, a study on the possibilities of FO replacement by a blend of VO (linseed oil, corn oil, and olive oil) showed that total substitution was feasible without negative effects on growth performance or whole-body composition [36]. Considering that camelina is an alternative oilseed crop with a markedly low environmental impact and its oil has a high unsaturated fatty acid content, the present work aimed to know the effects of partial or total replacement of FO with CO on the survival rate, growth performance, and whole-body composition of juvenile tench.

## 2. Materials and Methods

### 2.1. Ethics Statement

According to Spanish law (RD 53/2013) and an EU directive (2010/63/EU), the experiments conducted in this study were approved by Ethics Committee of the University of León (reference ULE_016_2015). Fish health, welfare, and environmental conditions in the experimental tanks were checked twice daily by visual observation of animal behavior. Water quality parameters, such as oxygen saturation, temperature, and water flow, were measured periodically (see Section 2.2.). Animals used to analyze whole-body composition were euthanized with an overdose of tricaine methanesulfonate (MS222, Ortoquímica S.L., Barcelona, Spain) at 300 mg L^−1^ by prolonged immersion. At the end of the experiment, remaining animals were transported to the fish farm from where breeders came for further maintenance.

### 2.2. Fish, Facilities, and Experimental Procedure

Tench larvae were obtained by hatching using artificial reproduction techniques [37] and reared in outdoor tanks. After four months, 540 juvenile tench from a homogenous pool were randomly distributed into groups of 30 fish in 18 fiberglass tanks (0.5 × 0.25 × 0.25 m) containing 25 L of water to obtain replicates corresponding to the different feeding treatments. Prior to distribution, 100 juveniles were anesthetized with MS-222 to measure initial total length (TL) and weight (W). TL was measured with a digital caliper (to the nearest 0.01 mm) and W was determined with a precision balance (to the nearest 0.001 g), after removing excess water with tissue paper. Values of 31.48 ± 0.49 mm and 0.33 ± 0.18 g (mean ± SEM) were obtained. Total biomass of each tank was weighted. Following a monofactorial design, diet was the experimental factor with 3 replicates per level of treatment.

At the beginning and the end of the experiment, a sample of 10 g of juveniles was taken from the same pool to determine proximate chemical composition and fatty acid profile of whole body (see Section 2.4.).

Artesian well water was supplied in an open system (flow-throughout) and each tank had a water inlet (inflow 0.30 L min^−1^), outlet (provided with a 250 µm mesh filter), and light aeration. Measures of the incoming water quality, ammonia, nitrites, hardness, and total suspended solids were performed once a week with a spectrophotometer: HACH DR2800 (Hach Lange GMBH, Vigo, Spain). Dissolved oxygen in tanks was measured with a multimeter: HACH HQ30d (Hach Lange GMBH, Vigo, Spain). Water temperature was recorded twice a day. A photoperiod of 16 h light: 8 h dark was maintained. Tanks were cleaned of feces and uneaten feed every two days. The experiment lasted for 90 days.

### 2.3. Diets and Feeding

Six diets (nearly isonitrogenous and isoenergetic) with different replacement levels of cod liver oil by camelina oil were formulated: 0% (control), 20%, 40%, 60%, 80%, or 100%. The ingredients were ground in a rotary BRABENDER mill (Brabender GmbH & Co. KG, Duisburg, Germany), mixed in a STEPHAN UMC5 mixer (Stephan Food Service Equipment, Hameln, Germany), and extruded using a stand-alone extruder BRABENDER KE19/25D (Brabender GmbH & Co. KG, Duisburg, Germany) at a temperature range between 100 °C and 110 °C. Pellets (1 mm diameter) were dried for 24 h at 30 °C and after receiving a coating of cod liver oil and/or camelina oil. Formulation of the practical diets is presented in Table 1. Fish were fed manually three times a day (at 10:00, 14:00, and 18:00 h) to apparent satiation.

### 2.4. Chemical Analysis of Diets and Fish

Juveniles fasted for 14 h before sampling. Samples of diets and juveniles were stored at −30 °C at the beginning and end of the experiment. Due to the small size of juveniles, proximate composition and fatty acid profile were made on whole body. Analyses were performed in duplicate by Analiza Calidad laboratory (Burgos, Spain) following Commission Regulation (EC) 152/2009. Moisture was determined by drying at 105 °C, crude protein according to the Kjeldahl method, crude lipid by extraction with light petroleum and further distillation, ash by calcination at 550 °C, and gross energy according to EU regulation 1169/2011. The content of carbohydrates was calculated by subtracting moisture, protein, lipid, and ash content from the wet weight.

Fatty acid profiles were determined by hydrophilic interaction chromatography (HPLC) using the AccQTag method from Waters (Milford, MA, USA).

From whole-body fatty acid content (g kg^−1^), the following indices were calculated:

ΣPUFA/ΣSFA = sum of polyunsaturated fatty acids × sum of saturated fatty acids^−1^

EPA + DHA = eicosapentaenoic acid + docosahexaenoic acid

Σn − 6/Σn − 3 = sum of omega 6 fatty acids × sum of omega 3 fatty acids^−1^

### 2.5. Data Collection and Statistical Analysis

Juvenile tench behavior was observed and registered after cleaning, feeding, and measuring the water quality parameters.

In order to have information about the growth performance evolution, a sample of 15 fish per tank (45 per treatment, 50% of total) was anesthetized to be individually weighed and measured every thirty days. TL and W were measured as described in Section 2.1., and then juveniles were gently returned to their respective tanks.

After 90 days, surviving fish were anesthetized and observed one by one using a magnifying glass to detect externally visible deformities affecting spinal axis, operculum, mouth, and tail fin. W and TL were measured individually, and total biomass per tank was weighed. The following indices were calculated:

Survival rate (%): 100 × (final number of juveniles x initial number of juveniles^−1^)

Specific growth rate, SGR (% d^−1^) = 100 × [(ln final W − ln initial W) × days^−1^)].

Fulton’s coefficient or condition factor, K = 100 × (W × TL − 3).

Biomass gain, BG (g) = final biomass per tank – initial biomass per tank.

Feed conversion ratio, FCR = total amount of feed fed per tank × BG^−1^ [38].

All treatments were replicated three times, and the experimental unit was a tank.

After confirmation of normality and homogeneity of variance, statistical analysis of growth performance and whole-body composition data were done by one-way analysis of variance (ANOVA) and polynomial contrasts with the SPSS16.0 computer program (SPSS, Chicago, IL, USA). Significant differences between means were estimated by Tukey’s multiple range test. *p* < 0.05 was used for rejection of null hypothesis.

## 3. Results

The mean values of water quality were pH, 7.5; hardness, 5.3 German degrees (calcium, 32.8 mg L^−1^); total suspended solids, 34.0 mg L^−1^; dissolved oxygen, ranged between 5.9 and 7.4 mg L^−1^; ammonia, <0.10 mg L^−1^; and nitrites, <0.010 mg L^−1^. The water temperature was 25 ± 1 °C.

The proximate composition and fatty acid profiles of the diets are in Table 2 and Table 3, respectively. It is remarkable that, compared to the control diet, the fatty acid profile was affected by the replacement of FO with CO, showing that some essential fatty acids, such as ALA (18:3n − 3), LA (18:2 − n6), and arachidonic acid (20:4n6), tended to increase as CO did. Other fatty acids such as palmitoleic (14:0), erucic (22:1n − 9), EPA (20:5n − 3), DHA (22:6n − 3), and dihomo-ɣ-linolenic (20:3n − 6) showed a reduction when CO was included.

From observations of juvenile tench behavior, the diets were equally ingested throughout the experiment, independent of the amount of CO included.

Table 4 shows the TL and W of juvenile tench after 30 and 60 days of the experiment. There were no significant differences between the juveniles fed different diets after 30 (mean value range: 40.55–43.55 mm and 0.98–1.19 g) and 60 days of the experiment (mean value range: 51.43–54.43 mm and 2.00–2.68 g). Mean TL increases of 34.1% and 69.3% were achieved after 30 and 60 days, respectively. From the initial weight, the mean W was 3.4 times higher after 30 days while at 60 days was almost seven times higher.

Growth performance values and indices over 90 days are shown in Table 5. No mortality occurred throughout the experiment. There were no differences in TL, W, SGR, FCR, K, or BG between the juveniles fed different diets, showing the feasibility of full FO replacement without adverse effects on growth performance. Fish with externally visible deformities were not observed.

The proximate composition and whole-body fatty acid profile of the juvenile tench at the beginning and end of the experiment are summarized in Table 6 and Table 7. The diets affected juvenile composition, so lipid content was significantly lower in fish fed 40% CO and higher levels of FO replacement.

After 90 days, the content of ALA in whole-body juveniles increased as FO substitution did, which was significantly higher with respect to the control diet from the 40% FO replacement diets. A significant increase of 20:4n − 6 (arachidonic acid) in the juveniles fed diets with 60% CO and higher was evidenced. The content of 22:1n − 9 (erucic acid) decreased in the juveniles fed diets with 80% CA and higher. No differences in ΣSFA, ΣMUFA, ΣPUFA, Σn − 3, or Σn − 6 were found.

Table 8 includes some indices commonly used to determine the nutritional value of fish as human food. The ΣPUFA/ΣSFA and Σn − 6/Σn − 3 ratios showed a trend for a linear increase with dietary CO whereas the EPA + DHA showed the opposite tendency. So, compared to the control diet, EPA + DHA content was significantly lower in the juvenile tench fed with a 100% CO diet, and the Σn − 6/Σn − 3 ratio was significantly higher in the juvenile tench fed with 80% and 100% CO diets.

## 4. Discussion

Palatability is considered one of the primary characteristics of aquafeeds to guarantee optimum feed intake and to reduce uneaten feed. Kasumayan and Døving [39] stated that fish are attracted to compounds that are water soluble, amphoteric, nonvolatile, and nitrogen-containing with a low molecular weight. Considering this, lipids seem to play a minor role in the palatability of aquafeeds [40]. However, the high content of glucosinolates in C. sativa could be a disadvantage, as these substances can be responsible for a bitter or sharp taste [41,42]. In fact, Ofori-Mensah et al. [17] in gilthead sea bream, Betancor et al. [26] in European sea bass, and Hixon and Parrish [18] in Atlantic cod reported a reduction of growth when fishe were fed diets including CO, which attributed to a reduction in feed intake due to a decrease of palatability. In our experiment, tench juveniles equally accepted all diets, leading us to consider that the flavor and taste were acceptable, even when FO was totally replaced.

Essential fatty acid (EFA) requirements in freshwater species can generally be satisfied by C:18 PUFA, as they are able to convert LA and ALA into n − 6 and n − 3 LC PUFAs. As it was expected, the inclusion in diets of CO provided increased amounts of ALA and LA whereas DHA and EPA content decreased (see Table 3). Nevertheless, total FO replacement by CO was feasible without negative effects on the growth performance in juvenile tench, showing that EFA requirements were fully covered. This study is consistent with those reported on Atlantic salmon [22,25] and in other freshwater species, such as rainbow trout [20] and Nile tilapia [21], where total FO replacement by CO had no detrimental impact on growth parameters. However, only partial FO replacement by CO has been feasible in marine species, exhibiting poorer growth with total substitution [16,17,18,19] except for the studies performed on juvenile Atlantic cod by Morais et al. [43] and juvenile red sea bream by [29]. In the latter, the authors hypothesized that the requirement for n − 3 LC PUFA may have likely been fulfilled through EPA and DHA from lipids provided by fishmeal included in the experimental diets, allowing for a good growth response even when fish oil was fully replaced by vegetable oils.

It is widely recognized that final fish composition reflects feed consumed, and thus, diets could alter the nutritional properties and quality of fish as human food [6]. Substitution of FO with CO generally does not affect the total lipid content in muscles or carcasses in either marine fish [16,17,27,43] or some salmonid species [20,21,22,23,25]. These findings are in disagreement with our results where, compared to the control diet, whole-body lipid content decreased significantly for the tench juveniles fed diets that included 40% and higher CO levels. In the same way, Mzengereza et al. [29] and Maldonado-Othón et al. [19] reported a decrease in lipid content in muscle when camelina oil was included in the diet. In studies with juvenile golden pompano (*Trachinotus ovatus*) and juvenile tench, Guo et al. [44] and Sáez-Royuela et al. [36], respectively, found an association between body lipid content and dietary ALA/LA so that the lowest body lipid content coincided with the highest ALA/LA ratio. In our experiment, the same relationship was found, and the high lipid deposition corresponded to juveniles fed the control diet (ALA/LA = 0.25) whereas juveniles fed 100% CO (ALA/LA = 1.14) had lower lipid whole-body content.

Fish products are highly recommended for human consumption as a primary dietary source of n − 3 LC-PUFA, which plays important roles in the growth and maintenance of health, preventing chronic cardiovascular diseases, diabetes, cancer, and age-related degenerative diseases [45]. A common negative effect in all studies where dietary CO was included was the reduction of EPA and DHA in whole-body fish or fillets. In this experiment, although EPA and DHA content in diets decreased significantly as CO increased, no differences in n − 3 LC-PUFA in whole-body juveniles were found. This confirms the results of Garrido et al. [46] about the high capacity of tench to biosynthesize n − 3 LC-PUFA from ALA.

In agreement with reports on Atlantic cod [18,46], gilthead sea bream [16,17], red sea bream [29,35], and Nile tilapia [21], the inclusion of CO led to a higher content of ALA in diets, which in turn determined an increase of this fatty acid in whole-body tench. Taking into account that the human dietary intake of ALA inversely correlates with cardiovascular disease and cancer risk [47], CO seems to have improved the nutritional quality of tench.

One of the most common indices for evaluating the nutritional value of human foods is ΣPUFA/ΣSFA. The recommended values have a wide range between 0.50 and 1.62 depending on the fish species considered [48], and the higher the better. In this experiment, this ratio varied between 0.75 and 0.92, corresponding to the control diet and 100% CO diet, respectively. These values were similar to those obtained by Sáez-Royuela et al. [36] in juvenile tench fed diets with different substitutions levels of FO with a blend of vegetable oils. Linhartovà et al. [49] reported higher values in the flesh of commercial-sized tench either under extensive culture (average weight 210 g, ΣPUFA/ΣSFA: 1.32) or semi-intensive conditions (average weight 680 g, ΣPUFA/ΣSFA: 1.10). In this study, the values reached of this index were over 0.45, the minimum limit for foods regarded as undesirable for human health due to their potential to increase blood cholesterol levels [50].

EPA + DHA is a nutritional index recognized worldwide, and, according to FAO [51], the recommended minimum intake for optimal adult health is 0.25 g per day^−1^. In tench fillets, Linhartovà et al. [49] found values of EPA + DHA of 3.19–3.75 g kg^−1^ in fish cultured under semi-intensive or extensive conditions, respectively. In our study, the values tend to decrease as CO inclusion increased and were significantly lower when FO was totally replaced (3.16 g kg^−1^ in 100% CO vs. 6.07 g kg^−1^ in the control diet). The higher values obtained in our study, except for juveniles fed 100% CO, could be attributed not only to feeding conditions but also to the different sizes of the fish. The juvenile tench only received extruded diets, leading to a higher lipid deposition than the one reported by Linhartovà et al. [49] and, as a consequence, a higher EPA + DHA content. It is also necessary to consider that, due to the small size of the juvenile tench at the end of the experiment, our data were obtained from whole-body.

Σn − 6/Σn − 3 is a useful indicator not only to compare the relative nutritional value of different fish species but also to establish a healthy diet in humans. Simopoulos [52] stated that lower ratios are more desirable in reducing the risk of many chronic diseases of high prevalence in Western societies as well as in developing countries. According to Strobel et al. [53], this ratio varies significantly interspecifically, as it is higher in freshwater species, and depends on whether the fish are wild or come from aquaculture. In general, wild fish tend to have lower Σn − 6/Σn − 3 ratios compared to farmed fish, which are associated with higher amounts of n − 6 PUFA from the vegetable oils used as an alternative to fish oil in aquafeeds [53]. Except for the control diet, the values obtained in this study, between 0.42 and 0.52, were similar to the those reported in tench reared in extensive culture (Σn − 6/Σn − 3 = 0.47) where only natural food was available [49]. With respect to this index, camelina oil could be considered a good substitute for FO, as the nutritional value of intensively farmed tench was unaffected.

Due to the scarce information on nutritional indices values for tench in the scientific literature, comparison with our results should be interpreted with carefulness, mainly because of the small size of the animals and differences in feeding conditions. Thus, further research on the effects of a CO dietary lipid on the nutritional and sensory quality of tench reared to commercial size should be performed.

## 5. Conclusions

A total replacement of FO with CO did not affect the survival and growth performance of juvenile tench. Compared to the control diet, the whole-body lipid content was significantly lower in juveniles fed 40% and higher CO diets. There were no significant differences in the whole-body content of n − 3 LC-PUFA whereas an increase of ALA was evidenced in tench fed 40% and higher CO diets. Nutritional indices such as ΣPUFA/ΣSFA, DHA+EPA, or Σn6/Σn3 were nutritionally adequate and healthy for human consumption.

## Figures and Tables

**Table 1 animals-12-03362-t001:** Formulation of the practical diets (g kg^−1^ diet, wet basis) with different replacements of cod liver oil (FO) with camelina oil.

	FO Replacement (%)					
Ingredients (g kg^−1^)	0	20	40	60	80	100
Fish meal ^1^	645	645	645	645	645	645
Corn meal ^2^	166	166	166	166	166	166
Dried *Artemia* cysts ^3^	100	100	100	100	100	100
Carboxymethyl *cellulose* ^4^	30	30	30	30	30	30
Camelina oil ^5^	0	4	8	12	16	20
Cod liver oil ^6^	20	16	12	8	4	0
L-ascorbyl-2-monophosphate-Na ^7^	5	5	5	5	5	5
*Dicalcium phosphate* ^7^	10	10	10	10	10	10
Choline chloride ^7^	3	3	3	3	3	3
Soy lecithin ^8^	10	10	10	10	10	10
Sodium chloride ^9^	1	1	1	1	1	1
Mineral and Vitamin premix ^10^	10	10	10	10	10	10

^1^ Skretting España S.A., Ctra. de la Estación s/n 09,620 Cojóbar. Burgos. España. ^2^ Adpan Europa S.L., ES-33186 El Berrón. Siero. Asturias. Spain. ^3^ INVE Aquaculture Nutrition. Hoogyeld 91. Dendermonde. Belgium. ^4^ Helm Iberica S.A., ES-28108 Alcobendas. Madrid. Spain. ^5^ Camelina Company. Camino de la Carrera, 11–11, 28,140 Fuente el Saz, Madrid. ^6^ Acofarma distribution S.A., ES-08223 Terrassa. Barcelona. Spain. ^7^ Cargill, ES-28720 Colmenar Viejo. Madrid. Spain. ^8^ Biover N.V., Monnikenwerve 109. B-8000 Brugge. Belgium. ^9^ Unión Salinera de España S.A., ES-28001 Madrid. Spain. ^10^ Provides mg kg^−1^ premix: inositol, 50,000; thiamin, 500; riboflavin, 800; niacin, 5000; pyridoxine, 1500; pantothenic acid, 5000; biotin, 150; folic acid, 3500; cyanocobalamin, 5; retinol, 2400; α-tocopherol, 30,000; cholecalciferol, 6.25; naphthoquinone, 5000; butylated hydroxytoluene, 1500; MgSO_4_-7H_2_O, 300,000; ZnSO_4_-7H_2_O, 11,000; MnSO_4_-H_2_O, 4000; CuSO_4_-5H_2_O, 1180; CoSO_4_, 26; FeSO_4_-7H_2_O, 77,400; KI, 340; Na_2_SeO_3_, 68.

**Table 2 animals-12-03362-t002:** Proximate composition (g kg^−1^ diet, wet basis) of the practical diets with different replacements of cod liver oil (FO) with camelina oil (CO).

FO Replacement (%)						
Proximate Composition (g kg^−1^)	0	20	40	60	80	100
Moisture	65.8 ± 3.1	60.5 ± 2.2	67.7 ± 3.4	67.3 ± 2.1	68.3 ± 1.5	64.2 ± 2.5
Crude protein	509 ± 4.0	508 ± 2.0	509 ± 5.0	509 ± 3.0	504 ± 2.0	510 ± 3.0
Crude lipid	99.1 ± 1.8	99.3 ± 1.8	99.4 ± 1.7	100.0 ± 2.0	99.8 ± 2.1	99.6 ± 2.2
Carbohydrates	196.1 ± 3.2	202.0 ± 3.5	196.1 ± 2.6	193.3 ± 0.5	199.4 ± 3.8	196.5 ± 3.4
Ash	130.0 ± 1.8	130.2 ± 1.9	128.1 ± 1.5	126.6 ± 2.0	130.0 ± 2.2	129.7 ± 1.8
Gross energy (MJ kg^−1^)	15.3 ± 0.1	15.2 ± 0.1	15.1 ± 0.1	15.4 ± 0.1	15.3 ± 0.2	15.3 ± 0.1

Values are mean ± standard deviation (SD).

**Table 3 animals-12-03362-t003:** Fatty acid profile (% of total lipid content) of the practical diets with different replacements of cod liver oil (FO) with camelina oil (CO).

FO Replacement (%)						
	0	20	40	60	80	100
14:0	4.17 ± 0.01	4.02 ± 0.06	3.82 ± 0.19	3.46 ± 0.0.18	3.16 ± 0.21	2.86 ± 0.05
15:0	0.52 ± 0.06	0.50 ± 0.04	0.49 ± 0.03	0.47 ± 0.03	0.44 ± 0.03	0.43 ± 0.06
16:0	17.14 ± 1.20	16.65 ± 0.93	16.25 ± 0.96	15.80 ± 0.82	15.18 ± 0.78	14.90 ± 0.75
17:0	0.47 ± 0.03	0.46 ± 0.02	0.44 ± 0.03	0.44 ± 0.04	0.41 ± 0.02	0.40 ± 0.03
18:0	2.98 ± 0.12	2.92 ± 0.14	2.92 ± 0.13	2.90 ± 0.15	2.76 ± 0.03	2.92 ± 0.11
20:0	0.10 ± 0.02	0.21 ± 0.06	0.28 ± 0.07	0.39 ± 0.08	0.49 ± 0.10	0.60 ± 0.10
24:0	0.30 ± 0.04	0.28 ± 0.02	0.27 ± 0.03	0.27 ± 0.04	0.26 ± 0.01	0.25 ± 0.03
14:1	0.55 ± 0.06	0.55 ± 0.05	0.53 ± 0.03	0.51 ± 0.03	0.45 ± 0.01	0.44 ± 0.02
16:1	8.16 ± 0.70	7.92 ± 0.60	7.44 ± 0.50	7.08 ± 0.40	6.29 ± 0.30	5.85 ± 0.20
17:1	0.12 ± 0.01	0.12 ± 0.02	0.12 ± 0.01	0.11 ± 0.01	0.10 ± 0.01	0.09 ± 0.01
18:1n − 9	26.8 ± 1.70	25.59 ± 1.80	24.95 ± 1.60	24.57 ± 1.50	23.73 ± 1.40	23.63 ± 1.40
20:1	6.24 ± 0.40	6.32 ± 0.30	6.32 ± 0.32	6.63 ± 0.44	6.88 ± 0.40	7.29 ± 0.61
22:1n − 9	0.61 ± 0.03	0.57 ± 0.03	0.51 ± 0.04 ^c^	0.48 ± 0.01	0.41 ± 0.03	0.32 ± 0.01
24:1	0.73 ± 0.01	0.66 ± 0.04	0.61 ± 0.03	0.64 ± 0.04	0.64 ± 0.04	0.64 ± 0.03
18:2-n − 6	6.63 ± 0.30	7.09 ± 0.50	8.17 ± 0.42	9.24 ± 0.18	10.40 ± 0.55	11.67 ± 0.67
18:3n − 6	< 0.05	0.07 ± 0.01	0.37 ± 0.03	0.37 ± 0.03	0.32 ± 0.01	0.31 ± 0.03
18:3n − 3	2.96 ± 0.13	5.15 ± 0.14	7.29 ± 0.22	9.12 ± 0.31	11.17 ± 0.42	13.31 ± 0.46
20:3n − 66	0.14 ± 0.01	0.12 ± 0.01	0.10 ± 0.01	<0.05	<0.05	<0.05
20:4n − 6	0.16 ± 0.02	0.23 ± 0.03	0.35 ± 0.03	0.36 ± 0.03	0.43 ± 0.06	0.50 ± 0.06
20:5n − 3	8.33 ± 0.31 ^a^	7.77 ± 0.34	7.11 ± 0.38	6.61 ± 0.34	6.18 ± 0.31	5.19 ± 0.27
22:6n − 3	8.97 ± 0.80 ^a^	8.28 ± 0.75	7.37 ± 0.72	6.62 ± 0.63	6.44 ± 0.61	5.02 ± 0.51
ΣSFA ^1^	25.68 ± 1.00	25.03 ± 0.91	24.47 ± 0.89	23.73 ± 0.87	22.74 ± 0.79	22.36 ± 0.75
ΣMUFA ^2^	42.96 ± 1.10	41.78 ± 0.92	40.48 ± 0.87	39.99 ± 0.84	38.65 ± 0.83	38.27 ± 0.80
ΣPUFA ^3^	27.19 ± 0.94	28.71 ± 1.03	30.71 ± 1.02	32.22 ± 1.22	34.94 ± 1.09	36.00 ± 1.09
ALA/LA ^4^	0.48 ± 0.001	0.73 ± 0.05	0.89 ± 0.01	0.99 ± 0.01	1.07 ± 0.001	1.14 ± 0.001

Values are mean ± standard deviation (SD). Some minor fatty acids (<0.05%) are not shown. ^1^ Total saturated fatty acids ^2^ Total monounsaturated fatty acids ^3^ Total polyunsaturated fatty acids ^4^ Linolenic acid/linoeic acid. Means in the same row with different superscripts are significantly different (*p* < 0.05).

**Table 4 animals-12-03362-t004:** Growth performance of juvenile tench fed practical diets with different replacements of cod liver oil (FO) with camelina oil (CO) over 30 and 60 days.

											Polynomial Contrasts	
FO Replacement (%) 0			20	40	60	80	100	SEM	ANOVA	Linear	Quadratic	Cubic
30 days	TL ^1^ (mm)	41.86	43.55	42.41	43.16	40.55	42.2	0.64	0.861	0.497	0.342	0.971
	W ^2^ (g)	1.18	1.17	1.11	1.19	1.09	0.98	0.05	0.886	0.342	0.597	0.683
60 days	TL ^1^ (mm)	54.23	51.43	52.38	54.34	54.11	53.3	0.66	0.808	0.705	0.736	0.216
	W ^2^ (g)	2.68	2.00	2.22	2.30	2.29	2.32	0.09	0.424	0.630	0.207	0.169

Values are mean and pooled standard error of the mean (SEM). ^1^ Total length. ^2^ Weight.

**Table 5 animals-12-03362-t005:** Growth performance of juvenile tench fed practical diets with different replacements of cod liver oil (FO) with camelina oil (CO) over 90 days.

	FO Replacement (%)						Polynomial Contrasts				
	0	20	40	60	80	100	SEM	ANOVA	Linear	Quadratic	Cubic
TL ^1^ (mm)	66.54	67.01	66.85	68.58	68.64	66.83	0.439	0.633	0.410	0.329	0.284
W ^2^ (g)	4.78	4.64	4.47	4.79	4.68	4.59	0.089	0.933	0.808	0.820	0.463
SGR ^3^ (% day^−1^)	2.78	2.44	2.78	2.84	2.85	2.81	0.060	0.404	0.265	0.903	0.159
K ^4^	1.44	1.41	1.39	1.37	1.37	1.45	0.009	0.011	0.166	0.432	0.197
FCR ^5^	1.20	1.26	1.26	1.21	1.22	1.22	0.013	0.674	0.879	0.382	0.188
BG ^6^ (g)	137.75	130.39	124.00	136.68	131.53	131.14	1.074	0.812	0.791	0.571	0.377

Values are mean and pooled standard error of the mean (SEM). ^1^ Total length; ^2^ Weight; ^3^ Specific growth rate = 100 × [(ln final body weight − ln initial body weight) × days) ^−1^]; ^4^ Condition factor = 100 × (body weight × [(body length^3^)^−1^]; ^5^ Feed conversion ratio = total amount of feed supplied per tank × BG^−1^; ^6^ Biomass gain = final biomass per tank – initial biomass per tank.

**Table 6 animals-12-03362-t006:** Proximate composition (g kg^−1^, wet basis) of the whole body of juvenile tench fed practical diets with different replacements of cod liver oil (FO) by camelina oil (CO).

		FO Replacement (%)						Polynomial Contrasts				
	Initial	0	20	40	60	80	100	SEM	ANOVA	Linear	Quadratic	Cubic
Moisture	768.7	763.6	768.6	759.5	774.3	760.3	773.8	3.46	0.44	0.64	0.81	0.72
Protein	147.2	159.9	145.7	145.8	145.3	149.1	151.4	1.80	0.15	0.22	0.03	0.89
Lipid	19.4	59.2 ^a^	56.1 ^ab^	48.0 ^b^	48.4 ^b^	38.5 ^c^	32.2 ^c^	2.39	<0.001	<0.001	0.001	0.10
Ash	24.2	22.5	19.1	18.1	22.6	22.5	18.2	0.70	0.06	0.47	0.93	0.60

Values are mean and pooled standard error of the mean (SEM). Means in the same row with different superscripts are significantly different (*p* < 0.05).

**Table 7 animals-12-03362-t007:** Fatty acid profile (% of lipids) of the whole body of juvenile tench fed practical diets with different replacements of cod liver oil (FO) with camelina oil (CO).

		FO Replacement (%)						Polynomial Contrasts				
	Initial	0	20	40	60	80	100	SEM	ANOVA	Linear	Quadratic	Cubic
14:0	3.29	2.65	2.62	2.57	2.41	2.23	2.23	0.06	0.06	0.005	0.68	0.35
15:0	0.45	0.37	0.37	0.37	0.40	0.36	0.34	0.01	0.55	0.42	0.22	0.50
16:0	16.50	20.53	21.07	20.07	19.49	19.39	19.95	0.33	0.79	0.32	0.68	0.41
17:0	0.33	0.24	0.24	0.27	0.27	0.25	0.23	0.01	0.44	0.88	0.08	0.59
18:0	2.78	2.77	2.71	2.61	2.65	2.58	2.58	0.04	0.70	0.17	0.96	0.88
20:0	0.14	0.18	0.21	0.22	0.25	0.27	0.29	0.01	0.17	0.01	0.97	0.98
24:0	0.23	0.16	0.18	0.15	0.17	0.14	0.17	0.01	0.55	0.72	0.74	0.28
14:1	0.39	0.33	0.35	0.35	0.35	0.32	0.26	0.01	0.21	0.08	0.06	0.66
15:1	0.14	0.14	0.11	0.11	0.12	0.11	0.10	0.01	0.39	0.11	0.61	0.19
16:1	12.3	11.52	11.66	11.35	10.74	10.54	10.82	0.30	0.92	0.36	0.89	0.59
17:1	0.13	0.63	0.62	0.63	0.72	0.67	0.58	0.01	0.07	0.96	0.03	0.03
18:1n − 9	27.82	33.98	33.20	32.79	31.36	31.79	32.53	0.41	0.57	0.19	0.29	0.57
20:1	2.23	3.30	3.20	3.50	3.63	3.91	3.91	0.10	0.39	0.05	0.90	0.64
22:1n − 9	1.09	1.01 ^a^	0.95 ^ab^	0.83 ^ab^	0.72 ^abc^	0.64 ^bc^	0.46 ^c^	0.06	0.007	<0.001	0.49	0.88
24:1	0.14	0.18	0.18	0.15	0.15	0.15	0.13	0.01	0.14	0.02	0.86	0.82
18:2n − 6	6.54	5.20	5.42	6.07	6.92	7.18	7.31	0.30	0.13	0.01	0.7118	0.52
18:3n − 6	0.49	0.19	0.20	0.20	0.23	0.21	0.20 ^b^	0.01	0.87	0.60	0.44	0.68
18:3n − 3	1.83	1.64 ^a^	2.41 ^ab^	3.40 ^bc^	4.36 ^cd^	5.11 ^de^	5.64 ^e^	0.43	<0.001	<0.001	0.33	0.39
20:3n − 6	0.47	0.31	0.32	0.31	0.37	0.38	0.41	0.01	0.06	0.06	0.41	0.53
20:4n − 6	0.12	0.15 ^a^	0.19 ^a^	0.27 ^ab^	0.34 ^bc^	0.39 ^bc^	0.41^c^	0.03	0.001	<0.001	0.35	0.28
20:5n − 3	2.39	3.61	3.51	3.42	3.48	3.24	3.21	0.08	0.79	0.20	0.91	0.91
22:6n − 3	10.06	8.73	7.63	8.11	8.25	7.77	6.49	0.28	0.32	0.09	0.44	0.18
ΣSFA ^1^	23.72	26.40	27.40	26.23	25.64	25.29	25.85	0.49	0.92	0.46	0.94	0.49
ΣMUFA ^2^	46.24	51.09	50.39	49.71	47.79	48.13	48.83	0.58	0.63	0.17	0.46	0.57
ΣPUFA ^3^	21.90	19.83	19.68	21.88	24.09	24.28	23.65	0.67	0.08	0.01	0.35	0.20
Σn − 6	7.76	5.85	6.13	6.85	7.86	8.16	8.33	0.34	0.09	0.008	0.70	0.49
Σn − 3	14.28	13.98	13.55	14.93	16.09	16.12	15.34	0.43	0.49	0.13	0.49	0.31

Values are mean and pooled standard error of the mean (SEM). Initial data were not included in the statistical analysis. Means in the same row with different superscripts are significantly different (*p* < 0.05). Some minor fatty acids (<0.05%) are not shown. ^1^ Total saturated fatty acids ^2^ Total monounsaturated fatty acids ^3^ Total polyunsaturated fatty acids.

**Table 8 animals-12-03362-t008:** Nutritional indices of the whole body of juvenile tench fed practical diets with different replacements of cod liver oil (FO) with camelina oil (CO).

	FO Replacement (%)						Polynomial Contrasts				
	0	20	40	60	80	100	SEM	ANOVA	Linear	Quadratic	Cubic
ΣPUFA/ΣSFA^1^	0.75	0.72	0.83	0.94	0.96	0.92	0.03	0.09	0.01	0.45	0.14
EPA + DHA ^2^	6.07 ^a^	6.25 ^a^	5.53 ^a^	5.67 ^a^	4.24 ^ab^	3.16 ^b^	0.35	0.01	0.001	0.05	0.86
Σn − 6/Σn − 3	0.42 ^a^	0.45 ^ab^	0.46 ^ab^	0.49 ^ab^	0.51 ^b^	0.52 ^b^	0.01	0.02	0.001	0.73	0.97

Values are mean and pooled standard error of the mean (SEM). Means in the same row with different superscripts are significantly different (*p* < 0.05). ^1^ Σ polyunsaturated fatty acids/Σ saturated fatty acids (g kg^−1^ wet weight). ^2^ Eicosapentaenoic acid + docosahexaenoic acid (g kg^−1^ wet weight).

## Data Availability

Data are available on the corresponding author’s request.

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
