# Peer review of "Fish Oil Replacement by Camelina (Camelina sativa L.) Oil in Diets for Juvenile Tench (Tinca tinca L.): Effects on Survival, Growth, and Whole-Body Fatty Acid Profile"

_animals, 2022, doi:10.3390/ani12233362_

Round 1

Reviewer 1 Report (Previous Reviewer 1)

Dear Authors,

thanks for your revision. I think the manuscript is improved. 

For next study, also on small fish, histological exams would be very interesting. 

Author Response

Thanks for your effort. We wil include histological exams in future research

Reviewer 2 Report (New Reviewer)

The manuscript deals with the total or partial replacement of fish oil with Camelina oil in the diet of Juvenile tench in terms of impact on survival, growth, and whole-body fatty acid. The experimental design is well executed following the standard methodology. The results are very interesting and the discussion part is concluded logically. Before I   could endorse this for publication, I do have some suggestions  to add a few parameters in order to improve the quality of  the manuscript.

Sensory quality such  as skin brightness, appearance transparency, flesh texture, flesh blood, and flesh odor is imperative to explore the potentiality of new alternative protein ingredients and is connected with consumer acceptance. The sensory evaluation of the raw and cooked flesh of the tench fed with fishmeal-free oil containing the different proportions of camelina oil might be an interesting parameter that should be included in this study.

Apart from this, microstructure changes in fish muscle could also shed important insight into the effect of alternative fish oil.

TBARS is extensively used to quantify the degree of second-stage lipid peroxidation of fish that yield aldehydes and ketones from the degradation of polyunsaturated fatty acids. This might be also considered for inclusion in your manuscript.

 line 62: Correct as  “due to “

Author Response

Point 1: The manuscript deals with the total or partial replacement of fish oil with Camelina oil in the diet of Juvenile tench in terms of impact on survival, growth, and whole-body fatty acid. The experimental design is well executed following the standard methodology. The results are very interesting and the discussion part is concluded logically. Before I   could endorse this for publication, I do have some suggestions  to add a few parameters in order to improve the quality of  the manuscript.

Sensory quality such as skin brightness, appearance transparency, flesh texture, flesh blood, and flesh odor is imperative to explore the potentiality of new alternative protein ingredients and is connected with consumer acceptance. The sensory evaluation of the raw and cooked flesh of the tench fed with fishmeal-free oil containing the different proportions of camelina oil might be an interesting parameter that should be included in this study.

Apart from this, microstructure changes in fish muscle could also shed important insight into the effect of alternative fish oil.

TBARS is extensively used to quantify the degree of second-stage lipid peroxidation of fish that yield aldehydes and ketones from the degradation of polyunsaturated fatty acids. This might be also considered for inclusion in your manuscript.

Answer: We want to thank your suggestions to improve the manuscript. The study was foccussed on the initial life stages of tench and, for this reason, the small size reached by of juveniles did not allow a evaluation of its sensory quality. As we stated in the manuscript, we will perform further research working with commercial size tench which will allow to consider not only the sensory evaluation, but also the other measures you have suggested. Unfourtunately we can not be able to give more information than the included actually in the MS. Your interesting recommendations are relevant in this research field and will considered them in future experiments

Point 2: line 62: Correct as  “due to “

Answer: It was corrected

This manuscript is a resubmission of an earlier submission. The following is a list of the peer review reports and author responses from that submission.

Round 1

Reviewer 1 Report

The article describes a trial in which camelina oil sobstitutes fish oil in the diet of juvenile tench (Tinca tinca). Despite the article is well written and interesting, I cannot understand how a trial like this, where a juvenile fish (line 123) is suppressed to determinate the chemical composition and fatty acid profile of the whole body, can be performed out of the umbrella of a permit for animal experimentation. Authors tell that the experiment lasted for 90 days (line 134) and in Table 5 they tell that the fish weight around 4-5 grams at the end of the experiment, so the trial cannot have considered a zoothecnical cycle.

You can read the European Directive 2010/63/EU on the protection of animal used for scientific purpose, at this link:

https://eur-lex.europa.eu/LexUriServ/LexUriServ.do?uri=OJ:L:2010:276:0033:0079:en:PDF

Chapter 1, article 1.3 tells that the Directive is applied shall apply to live non-human vertebrate animals, including independently feeding larval forms.

Indeed, juvenile tench are included in this definition.

This is the Spanish law:

I don’t think that the approval of the Ethics Committee of the Leon University (line 134) is sufficient to have the permit to make the experiment, according to the law. However, I would ask the Authors to see this authorization and a detailed explanation from this Committee about the regularity of this trial according with the European Directive.

I am sorry if I don’t know well the Spanish law about animal protection.

Another point: authors analyzed juvenile fish but they did not show what is the impact of camelina oil on liver and intestinal structure. Commercial size of tench is around 200 g, so the fish need a lot of time to grow. How can the substitution impact on its life? This point should be discussed as a limitation of the study.

Reviewer 2 Report

The research was on the effect of dietary replacement of fish oil by camelina oil on the growth performance, fish biochemistry,and whole body fatty acid profile of juvenile tench. The research was scientifically designed. However, there are some suggestions.

1. Using One-way ANOVA analysis in the comparison of fatty acids profiles of diet (Table 3) is inappropriate since each diet can only be viewed as one sample. Each replicates used in the chemical analysis cannot be regarded as independent samples.

2.Since the n-6/n-3 level and the content of DHA+EPA in fish fed with 100% CO group was significantly lower than the other diet, it is not appropriate to give the conclusion in the abstract that the total replacement of FO by CO would not bring negative affect to the nutritional quality. In addition, the optimum replacement level should be added.

3.Table 3."20:3n-66" should be corrected to "20:3n-6"